# The “Bagno dell’Acqua” Lake as a Novel Mars-like Analogue: Prebiotic Syntheses of PNA and RNA Building Blocks and Oligomers

**DOI:** 10.3390/ijms26146952

**Published:** 2025-07-19

**Authors:** Valentina Ubertini, Eleonora Mancin, Enrico Bruschini, Marco Ferrari, Agnese Piacentini, Stefano Fazi, Cristina Mazzoni, Bruno Mattia Bizzarri, Raffaele Saladino, Giovanna Costanzo

**Affiliations:** 1Department of Ecological and Biological Sciences (DEB), University of Tuscia, 01100 Viterbo, Italy; valentina.ubertini@unitus.it (V.U.); eleonora.mancin@unitus.it (E.M.); saladino@unitus.it (R.S.); 2Institute for Space Astrophysics and Planetology—INAF, 00133 Rome, Italy; enrico.bruschini@inaf.it (E.B.); marco.ferrari@inaf.it (M.F.); 3Department of Biology and Biotechnology “C. Darwin”, Sapienza University of Rome, 00185 Rome, Italy; agnese.piacentini@uniroma1.it (A.P.); stefano.fazi@irsa.cnr.it (S.F.); cristina.mazzoni@uniroma1.it (C.M.); 4Water Research Institute, National Research Council (IRSA-CNR), Montelibretti, 00010 Rome, Italy; 5Institute of Molecular Biology and Pathology, National Research Council—IBPM-CNR, 00185 Rome, Italy

**Keywords:** Origin of Life, prebiotic syntheses, RNA world, RNA abiotic polymerization, Mars analogues

## Abstract

The ongoing exploration of planets such as Mars is producing a wealth of data to define habitable environments beyond the Earth. The inferred presence of neutral to alkaline aqueous fluids on Mars in its early history suggests that many potentially habitable environments existed on the planet. Terrestrial analogues with similar chemical and physical properties are being explored and characterized in order to assess their suitability for triggering the Origin of Life on Mars. Recently, a novel Mars analogue site has been identified in the Bagno dell’Acqua Lake, which is located in the island of Pantelleria in Sicily (Italy). We report here that microbialite from the Bagno dell’Acqua Lake acts as an efficient catalyst for prebiotic processes, starting from a ternary mixture of well-recognized chemical precursors, including ammonium formate, diaminomalonitrile, and alpha-amino acids. Under thermal conditions, significant amounts of building blocks of both RNA and PNA were obtained. Furthermore, samples of the water from the Bagno dell’Acqua Lake have been found to promote the polymerization of the H-form of 3′,5′-cyclic GMP, resulting in the generation of RNA oligomers of up to 15 units in length.

## 1. Introduction

Analogue sites on the Earth are places characterized by one or more features similar to those observed or inferred on planetary surfaces. Many analogue sites were identified on the Earth and their geology, geomorphology, mineralogy, chemistry, and biology were extensively characterized [1,2,3,4,5,6,7,8]. One interesting site is the Bagno dell’Acqua Lake on the island of Pantelleria. Pantelleria is a Pleistocene stratovolcano island that shows a complex geologic history characterized by peralkaline volcanism [9]. It is dominated by a rheomorphic unit zoned from pantellerite to trachyte and consists of falls, surges, and pyroclastic flows [9,10]. The surface area of the Bagno dell’Acqua Lake is characterized by alkaline waters (pH = 9) and Eh values indicating strongly oxidizing conditions [11]. The lake is fed continuously by thermal springs situated on its shores and by meteoric waters [12]. In the thermal spring area, the temperature ranges between 34 °C and 54 °C and the pH is about 6.2 [13]. Water–rock interaction and evaporation strongly contribute to the chemical composition of the lake water, resulting in high contents of Cl^−^ and SO_4_^2−^, being classified as the chloride–sulfate–alkaline type with Cl^−^ as the predominant species [13]. The lake water chemistry is characterized by the following general pattern: Na**^+^** > K**^+^** > Mg^2**+**^ > Ca^2**+**^ for cations and Cl^−^ > HCO^3−^ > SO_4_^2−^ > CO_3_^2−^ for anions; the chemical composition of major elements is in reference [14,15]. In situ studies reported carbonate and silica precipitates of microbial origin (silica stromatolites and silica gel) [15,16], observed/detected also on the surface of Mars. The physical characteristic of Bagno dell’Acqua makes it a possible Martian analogue, since there are depressions on Mars that have hosted endorheic lakes (e.g., Huygens Crater) [17]. Having hosted water in the past, these basins are of astrobiological interest because they may have preserved organic matter [18,19]. Endorheic basins or, more in general, ancient lakes on Mars are characterized by the presence of clay minerals (Fe, Mg, and Al-clays), sulfates, and carbonates. Clay minerals were found in the Jezero and Eberswalde craters [20]; sulfates were detected in the Jezero and Gusev craters [21,22]; carbonates were observed in Columbia Hills (the Phoenix landing site); and amorphous silica was found in the Gusev crater and Oxia Planum [23,24]. These mineralogical features are in good agreement with those of the Bagno dell’Acqua Lake sediments. The geological environment of the Pantelleria area is dominated by peralkaline rhyolites (pantellerites), trachytes, and minor alkaline basalts that show similarities with the Columbia Hills site and the Gusev crater on Mars [10]. In addition, the hydrothermal environment is consistent with those widely associated with Martian Noachian deposits [25]. All these physical characteristics reinforce the idea that Bagno dell’Acqua Lake is a Martian analogue site as early as originally proposed by [12] and recently confirmed by [26]. The sediments in the lake are characterized by an association of Mg-smectite and Ca-carbonate in addition to other primary and secondary minerals [11,26]. Microbialites from the eastern shore of the lake mostly consist of carbonate minerals (aragonite and calcite; 1−65%), feldspars (K-feldspar and plagioclase; 6−8%), and subordinate amounts of smectite (3%) and quartz (2%) [15]. The role of minerals in prebiotic chemistry is well-established and discussed in relation to catalysis in pristine chemical transformations [27]. Minerals are hypothesized to play a crucial role in the concentration and organization of organic molecules, facilitating the formation of complex and functional networks [28]. Carbonates, such as aragonite, are known to catalyze the synthesis of nucleobases from formamide under conditions that mimic early Earth environments [29]. In addition, clays promote a broader array of prebiotic syntheses from formamide and hydrogen cyanide (HCN) [30]. Clays not only support the formation of nucleobases but also enable the synthesis of carboxylic acids and sugars, which are ingredients for pre-metabolism [27]. Besides nucleobases and carboxylic acids and sugars, the scenarios for the Origin of Life suggest RNA (RiboNucleic Acid) as the key polymer for the catalysis and storage of genetic information (RNA world) [31,32,33]. The prebiotic syntheses of RNA in icy environments [34,35] and mineral surfaces [36,37] have been reported, with the challenge of the use of highly activated precursors [38]. As an alternative, mildly activated nucleotides may serve in the synthesis of short RNA sequences [39]. The 3′,5′-cyclic nucleotides form small RNAs, with 3′,5′-cyclic GMP (3′,5′-cyclic guanosine monophosphate) being the most reactive one, due to its unique stacking properties [40]. The polymerization of cGMP was revisited in dry form, as well as in a basic aqueous solution [41]. RNA and peptides may have co-evolved by RNA–peptide chimeras (Peptide Nucleic Acid, PNA) in which nucleobases are decorated with amino acid residues [42,43,44]. The selectivity of the decoration pattern was a consequence of molecular evolution, affording amino acids linked to the nucleobase in the exocyclic position, or, alternatively, at the N9, C-8, and N1 positions of the heterocyclic ring [45,46,47]. These latter derivatives show peptidase-like activity (prebiotic genotype–phenotype transfer machinery [48], since they can elongate peptides by Watson–Crick recognition between adjacent RNA molecules [49]. Unfortunately, few prebiotic syntheses are reported for the contemporary formation of RNA and PNA’s building blocks.

Here, we describe the effectiveness of microbialites from Lake Bagno dell’Acqua to catalyze the prebiotic synthesis of small and large organic molecules of relevance for the Origin of Life, including the complete set of RNA nucleobases and different PNA’s building blocks. The ternary mixture of diaminomaleonitrile (HCN tetramer; DAMN), ammonium formate (NH_4_^+^HCOO^−^), and alpha amino acids were selected as a prebiotic probe. This mixture is a sub-set of a largely possible general condition; it is not an advanced stage towards complexity: all the starting reagents are derivatives of formamide and HCN. Reactions were triggered by heating the ternary mixture in water in the presence of microbialite. Finally, the role of the Bagno dell’Acqua Lake environment in the prebiotic synthesis of RNA oligonucleotides from 3′,5′-cyclic GMP (hereafter abbreviated as cGMP) was studied, in order to evaluate the next step in molecular evolution towards chemical complexity.

## 2. Results

### 2.1. Role of Microbialite in Prebiotic Chemical Reactions

We selected a sample of microbialite collected and characterized from the eastern part of the Bagno dell’Acqua Lake (Figure 1, C1).

It contains a complex mineral assemblage: aragonite (65%wt), quartz (2%wt), smectite (3%wt), plagioclase (8%wt), K-felspar (6%wt), and halite (15%) [50]. The characterization of the microbialite sample is detailed in Appendix A. As a general procedure, ammonium formate **1** (0.092 mmol, 1.0 eq.), DAMN **2** (0.092 mmol, 1.0 eq.), and selected α-aminoacids **3a**–**3f** (glycine, Gly; L-alanine, Ala; L-valine, Val; L-serine, Ser; L-phenylalanine, Phe; and L-tyrosine, Tyr, respectively) (0.092 mmol, 1.0 eq.) were dissolved in distilled water (4 mL) in the presence of microbialite (5 mg) (microbialite was treated before use to remove absorbed organics as reported in the Materials and Methods section) and stirred at 50 °C for 24 h (reactions **A**–**F**). Microbialite was treated before use to remove absorbed organics as reported in the Materials and Methods section. The powder after treatment did not release any compounds after extraction with medium polar (ethyl acetate) and polar (methanol) organic solvents. The temperature of 50 °C was selected as corresponding to the actual temperature in the hydrothermal source of the Bagno dell’Acqua Lake (Figure 1, HP). The reaction of **1**, **2**, and **3a** (as a selected case) was also performed in the absence of microbialite as a reference. In this latter case, trace amounts of AICN (4-aminoimidazole-5-carbonitrile) and adenine were detected. After the separation of microbialite and freeze-drying, the crude was analyzed by gas chromatography associated with mass spectrometry (GC-MS) after standard derivatization procedures (details are in the Materials and Methods section). The structure of the reaction products was assigned based on a comparison of the molecular ion and fragmentation peaks with original commercial samples and by a comparison with data stored in a specific database (National Institute of Standards and Technology). The NIST database can easily discriminate between isomers by a comparison of different parameters, including the retention time, number, and variability of fragmentation peaks and the relative intensity between the major fragments. The representative *m*/*z* fragmentation peaks and relative peak abundances and retention time are reported in Appendix A. The original *m*/*z* fragmentation spectra of products and standard compounds are in Appendix A, respectively, while the original chromatogram of reaction **A** is reported in Appendix A as a selected example. The chemical structures and yield of the reaction products are reported in Table 1 and discussed in the following based on their specific relevance to the Origin of Life.

#### 2.1.1. Nucleobases and Analogues

Adenine **4**, guanine **5**, cytosine **6**, and uracil **7** were isolated as the complete set of RNA nucleobases, beside thymine **8** (Table 1; entries **1**–**5**). Nucleobase analogues, orotic acid **9**, and 2,6-diamino purine **10** were also detected in an appreciable amount. Orotic acid **9** is a key intermediate in the biosynthesis of pyrimidines [51], while 2,6-diamino purine **10** is a bioisoster of adenine which pairs efficiently with pyrimidine nucleobases by Watson–Crick interactions [52]. The formation of purines may be associated with DAMN chemistry [53]. DAMN isomerizes to DAFN (diaminofumaronitrile) and participates in reactions with HCN and single-carbon donors, such as formaldehyde, to yield AICN; this latter intermediate can further react with formamide and other prebiotic compounds to afford purines **4**–**5** and **10** (Figure 1) [54]. We have not studied in detail the mechanism of the DAMN to DAFN isomerization. It was probably triggered by both photons and heating under our experimental conditions. The generation, “in situ”, of formaldehyde from the HCN derivative is reported [55]. Note that AICN **11** was effectively detected in the reaction mixture (Table 1; reaction **B**). As an alternative, DAMN may be involved in pyrimidine ring-closure with isocyanate and formamide (produced from DAMN hydrolysis) to afford **6**–**9** (Figure 1) [56,57]. Compound **8** may be produced from **7** by a reaction with the formaldehyde and formic acid generated in situ [58,59]. As for the selectivity of the transformation, purine and pyrimidine nucleobases were obtained in a comparable yield, with 2,6-diamino purine **10** being isolated only in the case of reaction **A**. No specific relationships were evident between the type of the amino acid and the yield of the reaction products. As a general trend, nucleobases were isolated in the highest yield in the presence of serine (reaction **D**). Serine, alone among the common α-amino acids, spontaneously forms clusters in the solution, which are reported to facilitate prebiotic transformations [59].

#### 2.1.2. PNA’s Building Blocks

Five PNA’s building blocks decorated with amino-acid residues at both the N1 and N9 position of the purine ring were detected in different instances: N1-acetic acid adenine **12** and N9-acetic acid-2,6-diaminopurine **15** (reaction **A**), N1-(Ala)-adenine **13** (reaction **B**), N1-(Ser)-adenine **14** (reaction **D**), and N9-(Phe)-2,6-diaminopurine **16** (reaction **E**). PNA building blocks were not observed in reaction **F**; this result is in accordance with data previously reported on the low reactivity of aromatic amino acid in the synthesis of PNA’s nucleobases [45,46,47]. N9-substituted derivatives **15** and **16** are components of canonical PNA able to form a double helix with DNA [60], while the N1-substituted counterpart represents the skeletal variant in the PNA’s family. N1-substituted purines are characterized by significant biological activity, including the maturation-inducing hormone (1-Methyladenine) of starfish oocytes [61,62,63]. Compounds **12**–**14** are probably formed by a multicomponent process, involving the formation of AICN **11** as a key intermediate, followed by the reaction of **11** with ammonium formate **1** to afford intermediate **I** (not isolated) and successive ring-closure involving the participation of the amino acid (Figure 2; Pathway A). This hypothesis is in accordance with hte data previously reported in the synthesis of N1-substituted purines by a three-component reaction involving DAMN and trimethylortoformate [46]. As an alternative, compounds **15** and **16** may be produced by the initial formation of AMN (aminomalononitrile) from DAMN [64], followed by the condensation of AMN with ammonium formate and amino acid to afford intermediate **II** (not isolated in our case), and successive addition of **II** with generated in situ guanidine (Figure 2, pathway B). As for the selectivity of the transformation, N1- and N9-substituted purines were contemporarily obtained in the presence of Gly (compounds **12** and **15**, respectively); Ala and Ser afforded N1-substituted derivatives (compounds **13** and **14**), while Phe yielded an N9-substituted counterpart (compound **16**). As a general trend, the yield of PNA’s building blocks was lower than that of RNA nucleobases. In addition, the dimer **17** was tentatively assigned (reaction **A**) based on the molecular ion and characteristic pattern in the GC-MS fragmentation peaks. The presence of this compound suggests the possibility of the formation of complex fragments of PNA in which the nucleobases are linked together by the N-(2-aminoethyl)-glycine and ethylene diamine spacers.

#### 2.1.3. Peptides and Miscellanea

Four dipeptides, GlyGly **18**, AlaAla **19**, ValVal **20**, and SerSer **21**, were isolated. Dipeptides from Phe and Tyr were not observed. The formation of dipeptides from amino acids in the presence of HCN derivatives is reported and the mechanism of the condensation is described in detail (Figure 3). It involves the generation, in situ, of urea, a potent condensing agent, and the formation of intermediates **III** and **IV** (compounds **26** and **27** in the case of reaction **A**), followed by the nucleophilic addition of the second amino acid [65]. The occurrence of HCN chemistry in our experimental conditions was demonstrated by the isolation of oligomers of HCN, tentatively assigned as compounds **22**–**25**. An appreciable amount of N-carboxyamide (NCA) of glycine, compound **26**, was also detected in reaction **A**. NCAs are reactive intermediates in the synthesis of peptides [66], usually synthesized from amino acids and carbon dioxide [67] or from amino acids and isocyanic acid (Bücherer–Bergs carbamoylation) [68].

### 2.2. In Situ Abiotic Polymerization of cGMP

The polymerization of cGMP was studied by in situ experiments in the Pantelleria Bagno dell’Acqua Lake. Experiments were conducted during two different field campaigns (June 2023 and June 2024) with a total of 75 samples analyzed under different experimental conditions. The polymerization procedure and its evaluation consisted of successive steps, partly differing depending on the water content of the samples. For polymerization in the Bagno dell’Acqua Lake, the starting material consisted of the pure H-form of 3′,5′cGMP, dissolved in water, custom-made by BioLog LSI, following a procedure that avoids precipitation steps after the final HPLC (High-Performance Liquid Chromatography) analysis (as detailed in the Section 4.1). This point is of major relevance for the quality of prebiotic experiments, given that precipitations during the synthesis procedure by the manufacturer may introduce variability and the uncontrolled oligomerization of monomers and of short fragments, affording materials which are prone to misinterpretations. The logic for the use of this type of material and the necessary controls and precautions are detailed [40,69,70]. The cGMP monomers were dried in laboratory conditions (see Section 4.6) and suspended in situ either in lake water (filtered as described in Section 4.1) or in HPLC-grade pure water and incubated at 50 °C in the thermal spring (Figure 1; HP site), for varying time periods.

#### 2.2.1. Water Chemistry of the Bagno dell’Acqua Lake

The cGMP oligomerization reaction was studied under a multitude of experimental conditions. The temperature optimum was ca. 80 °C [40,69,70], in association to one drying step necessary for obtaining oligonucleotides longer than trimers [41,71]. Although it has been demonstrated that the presence of water in the reaction is almost poisonous for polymer formation and that only short (≤3 nt long) oligomers form in the solution [72], we wished to investigate whether the chemical and physical conditions present in the Bagno dell’Acqua Lake were nevertheless compatible with the possibility of in situ cGMP polymerization.

Bagno dell’Acqua is an alkaline, saline, endorheic lake (pH ca. 9). The lake is fed continuously by thermal springs situated on its shores and by meteoric waters. Consequently, the main processes governing the composition of the lake water are mixing between meteoric and thermal waters, as well as the dissolution of magma-derived volatiles, mainly CO_2_. Thermal springs and bubbling gases are located on the south-western area of the Bagno dell’Acqua Lake close to the perimeter lake road. In this area, and, in particular, in the thermal spring (Figure 1; HP site), the water shows temperatures ranging between 34 °C and 54 °C. The chemical composition of the major elements in the lake water samples is shown in references [13,14,15,73,74,75,76,77,78,79]. Lake waters taken near the shore are characterized by hydrochemical properties resembling the central lake, with a residual imprint of the spring fluids. The temperature, pH, and ion concentrations were only marginally lower than in the open equilibrated lake water.

#### 2.2.2. The Polymerization Reaction

Polymerization studies targeting 3′,5′ cyclic nucleotides place a special emphasis on identifying the conditions that are compatible with sustainable oligonucleotide production on longer time scales in a geochemical context relevant to the early Earth or in other planetary environments. To this purpose, the Bagno dell’Acqua Lake has been tested as an environmentally compatible site for cGMP abiotic oligomerization using the lake water as the solvent and one of its hot spring as the thermal energy source (Figure 1; HP site). cGMP monomers have been subject to a previous drying step, owing to the necessity of the preliminary formation of a pillared structure made of cGMP monomers [69], dissolved in water and incubated in situ at 50 °C. Figure 2 shows the PAGE (PolyAcrylamide Gel Electrophoresis) analysis of the oligomers formed as a function of time from the cGMP material suspended in the lake water (see Section 4.1). After incubation in the thermal spring for the indicated times, the reaction was immediately stopped by ethanol precipitation and quick freezing. The samples were terminally labeled and immediately analyzed by PAGE. The PAGE profiles were quantitatively evaluated by using the NIH image processing ImageJ 1.54K program.

Figure 3 shows the in situ polymerization control experiment in which the cGMP monomers were resuspended in HPLC-grade water (see Section 4.1) and treated as described above.

The comparison between panel (a) of Figure 2 and Figure 3 clearly indicates the catalytic effect of pH and inorganic ions on the rate of cGMP polymerization with the formation of oligomers to a length of at least 15 monomer units. Panels (b) and (c) of Figure 2 and Figure 3 show the length distribution of the oligomers produced as a function of time, while panels (d) and (e) of Figure 2 and Figure 3 show their average length, respectively. The longest period analyzed was 24 h, an amount of time sufficient for observing a wave-like behavior in terms of the amount and distribution of the oligomers as detailed in [41]. For each experiment, the profiles are not exactly repeatable as a precise function of time and as a quantity of polymer obtained, and for the time of onset of the degradation processes. However, the alternating character is constantly observed. This hints at the fact that the system is intrinsically complex and that one or more of its parameters are close to being stochastic. The re-execution of the in situ experiment the following year (2024 campaign) provided further confirmation of the results, thereby attesting to the experiment’s reproducibility (Appendix A).

The fact that soda lakes (in which Na**^+^** and (bi)carbonate are the dominant dissolved species) provide unique opportunities for the prebiotic formation of RNA has been demonstrated for activated ribonucleotides [80].

Despite the potential of certain ions dissolved in the lake (e.g., Na**^+^**) to inhibit cGMP polymerization [70], the data presented in Figure 2 (panel a) demonstrate that the multivariable effects of the Bagno dell’Acqua Lake environment act positively and that, likely, the role of most minerals is mainly to increase the hydrolytic stability of the polymerization products and the local concentration of cGMP monomers.

This aspect was further investigated by incubating another series of samples in the hot spring, in which cGMP was dissolved in Tris [Tris(hydroxymethyl)amino methane]-HCl buffered water at pH 9. As shown in Figure 4, the formation of oligomers with a maximum length of 13 units occurs after a period of one hour of incubation (although in smaller quantities than the oligomers formed in the lake water, as can be seen by comparing panels (b) and (c) in Figure 2 with panels (b) and (c) in Figure 4). However, in the absence of minerals, the newly formed RNA is rapidly degraded after three hours. A very marginal increase in dimers and trimers is again observed after 24 h.

#### 2.2.3. Effect of Sunlight on the Polymerization Reaction

It is hypothesized that environmental parameters, such as sunlight irradiation and thermal fluctuation, may be of relevance for the polymerization reaction. In order to test this hypothesis, a set of samples in which cGMP was suspended in lake water and placed in tubes that had been completely darkened was incubated in the hot spring. The result of the experiment is illustrated in Figure 5. The experiment revealed no polymerization products, with the exception of traces of oligomers up to five units in length, which did not increase with incubation time. It is evident that sunlight plays a pivotal role in this process; however, the precise mechanism through which it exerts its influence remains to be elucidated. The presence of trace amounts of anatase in the lake’s mineralogy [26] suggests the potential involvement of titanium dioxide (TiO_2_) in this process. Further investigation is necessary but it falls outside the scope of the present study.

## 3. Discussion

The Origin of Life likely occurred within environments that concentrated cellular precursors and enabled their co-assembly into cells. Soda lakes have been shown to concentrate precursors of RNA and membranes, such as phosphate, cyanide, and fatty acids [80]. The importance of RNA in protocells is highlighted by its ability to store and transmit genetic information, in addition to catalyzing reactions [31,32,33]. PNA’s alternative may improve the tool of genetic molecules useful for molecular evolution. We demonstrated that a ternary mixture of prebiotic precursors easily polymerizes under thermal conditions mimicking the Bagno dell’Acqua Lake in the presence of local microbialite to yield the complete set of RNA and DNA nucleobases, as well as a representative PNA building block. Among them, N9 and N1 purine derivatives decorated with different amino acids were isolated. These latter compounds were produced by a combination between the ternary mixture and reagents formed by the degradation, in situ, of the starting ones. A dimeric PNA derivative was also detected, suggesting a possible route to further improve molecular complexity. The prebiotic relevance of the transformation was enlarged by the contemporary synthesis of four dipeptides and of some HCN oligomers. The presence of the microbialite was essential, since the reaction repeated in the absence of the mineral was not effective. RNA oligomers were also obtained from 3′,5′-cGMP in the Bagno dell’Acqua Lake conditions, the first accredited Mars analogue site in Italy. The validity of the proposed simple polymerization mechanism [69] requires, in addition to the non-fastidious correct positioning of the reacting species obtained by purine stacking, favorable thermodynamics. Experiments performed in situ at the Bagno dell’Acqua Lake have enhanced our understanding of the abiotically driven polymerization mechanism of cGMP in several ways: (i) **pH and water chemistry**: The ring-opening polymerization reactions of cyclic phosphate and phosphonate esters in non-aqueous medium are well-known in the literature [81,82]. These reactions are commonly catalyzed by DBU (1,8-Diazabicyclo[5.4.0]undec-7-ene), suggesting that base-catalyzed ring-opening polymerization operates in this case. In the initial reports on the polymerization of 3′,5′ cGMP, the highest reaction yields were observed at pH 9, achieved with Tris-HCl buffering, suggesting a base-catalyzed reaction mechanism [69]. However, a comparison of the results of experiments carried out in Tris-HCl-buffered water and in lake water clearly demonstrates that pH alone is not sufficient for efficient polymerization. The intervention of dissolved salts in water is imperative, particularly in the protection of newly formed polymers from their hydrolytic degradation. The hydrolysis of RNA is a well-characterized process [83,84], and the hydrolytic protection by salts has already been demonstrated, for example, with condensed phosphate salts, like Na_4_P_2_O_7_ (tetrasodium pyrophosphate) or Na_5_P_3_O_10_ (pentasodium triphosphate), where the protection might be due to the relatively large Na^+^ cation concentration that enables phosphate clustering along the sugar-phosphate backbone [41]. While the contribution of individual ions to polymerization lies beyond the scope of the present work, the overall positive catalytic effect is evident, as is the role of the inorganic component in the stabilization of newly formed oligonucleotides over time. A protecting effect of the 3′,5′ cGMP H-form was also observed [41]. At a concentration of approximately 10 mM, cGMP is known to form aggregates [69] which are most likely stabilized by distinctly stable guanine–guanine stacking interactions. Since aggregate formation leads to phase separation and a concomitant decrease in the water activity along the backbone, the addition of cGMP may indeed enhance the hydrolytic stability of short oligoG sequences. It can, thus, be hypothesized that minerals could play a protective role in relation to degradation also by increasing the local concentration of cGMP molecules. (ii) **Sunlight**: It has been repeatedly shown that high-energy inputs, such as UV illumination, can generate organic molecules from H_2_O, CO_2_, and N_2_, and stimulate the formose reaction (where sugars and organic acids with four to seven carbon atoms are obtained from C1–C3 aldehydes and/or alcohols). Moreover, the CN-bond-containing molecules may have been formed in the primordial CO_2_- and N_2_-containing atmosphere, as well as in ice under the action of UV light [85,86]. Guanine, adenine, and hypoxanthine have been produced in UV-irradiated formamide solutions [87]. Moreover, the remarkable photostability exhibited by nucleosides and nucleotides suggests the possibility of a selection process that favors these compounds based on their inherent ability to resist UV light and high-energy radiation [88]. It is, therefore, conceivable that a process of abiogenesis may have occurred in sunlit geothermal pools, with abiotically polymerized cGMP being a plausible component of this process (in this instance, TiO_2_ could act as a catalyst, analogous to the process of carbon–carbon bond formation induced by solar light, as previously described in [89]).

In view of the foregoing observations, it is evident that the Bagno dell’Acqua Lake possesses considerable potential for the study of the Origin of Life. Further experiments will be able to take advantage of this environment towards an increasing molecular complexity.

## 4. Materials and Methods

### 4.1. Materials

Ammonium formate, DAMN, and aminoacids (Fluka; reagent grade) were used without further purification; analytical standards (**4**–**11**, **18**–**21**, **26**–**27**) have been purchased from Merck Rahway, NJ, USA (analytical grade, purity over > 99%) and used without further purification; compounds **12**–**16** have been synthesized and used as standard reference (synthetic procedure and analytical data are in Appendix A). GC-MS analyses were performed with a GC-MS with a Varian GC410-320MS (Palo Alto, CA, USA) equipped with a CP8944 column (WCOT-fused silica; film thickness, 0.25 μm; stationary phase VF-5 ms; Øί, 0.25 mm, length, 30 m). GC-MS analyses were also performed with a TRACE GC-MS with a Thermo Finnigan PolarisQ Ion Trap (San Jose, CA, USA) equipped with a CP8944 column (WCOT = fused silica; film thickness, 0.25 μm; stationary phase VF-5 ms; Øί, 0.25 mm, length, 30 m).

3′,5′ cGMP was obtained from BioLog LSI (Bremen, Germany) in acid form (3′,5′ cGMP, H +) as 1 mM solution in water, pH 3.4. The compound was custom-made and specially purified in order to guarantee (i) the maximal possible purity relative to the absence of adduct-forming cations (mostly Na**^+^**), and (ii) the absence of evaporation or precipitation steps in the course of the whole process. The process by the manufacturer BioLog consisted of the following: 735 µmol cGMP, Na^+^ (BioLog LSI Catalog No. B 004), MW 367.2 g/mol was dissolved in 490 mL deionized H_2_O (1.5 mM) and applied to a cation exchange column pre-equilibrated to the H^+^-form (5 L 0.5 M HCI, followed by 10 L H_2_O until neutral) at a flow rate of 5 mL/min. The strong cation exchanger Toyopearl SP-650M (TOSOH Bioscience, Stuttgart, Germany) (binding capacity: 0.15 meq/mL corresponding to a total of 73,500 meq) was used. This translated to a 100-fold excess of hydrogen over sodium, if 735 µmol cGMP, Na^+^ was applied to the cation exchanger. All product fractions with a concentration higher than 1 mM cGMP, H^+^ were pooled and stored at +4 °C. The column was regenerated with 5 L 0.5 M HCI and washed with 10 L H_2_O to remove all residual Na^+^ ions. Afterwards, the cGMP, H^+^ was applied again to the column to remove all remaining traces of Na^+^ ions. The resulting product fractions were pooled and filtered through a 0.2 µm membrane. The concentration was adjusted to 1 mM cGMP, H^+^ with deionized H_2_O and the solution was frozen at −26 °C. The purity of the final product was 99.67% (HPLC at 253 nm), as analyzed by the Provider.

Romil-SpS (Super Purity Solvent) water was used throughout for control samples. Lake water was filtered to 220 nm to remove contaminants, such as sediment and microbes.

G23 and G20 RNA oligonucleotides were purchased from Biomers.net (Ulm, Germany) and were provided in the standard dried form.

### 4.2. Preparation of Microbialite Powder

Then, 2 g of microbialite was gently crushed in agate mortar and the obtained powder was subjected to a thermal treatment (200 °C for 15 min) both to remove as much water as possible and to decompose thermally organic materials. Next, the microbialite powder was extracted with NaOH (1.0 mL; 0.1 N) and CHCl_3_-CH_3_OH (3.0 mL; 2:1 *v*/*v*) at room temperature. The recovered powder was again extracted with sulphuric acid (1.0 mL; 0.1 N) and CHCl_3_-CH_3_OH (3.0 mL; 2:1 *v*/*v*). The powder was recovered by centrifugation (6000 rpm, 10 min) and the supernatant phase was decanted. The powder after treatment did not release any compounds after extraction with medium polar (ethyl acetate) and polar (methanol) organic solvents.

### 4.3. Setting of the Reaction

In a round bottom flask, ammonium formate **1** (0.092 mmol, 1.0 eq.), DAMN **2** (0.092 mmol, 1.0 eq.), and a panel of selected α-aminoacid **3a**–**3f** (glycine, L-alanine, L-valine, L-serine, L-phenylalanine, and L-tyrosine) (0.092 mmol, 1.0 eq.) were dissolved in lake water (4 mL) in the presence of microbialite (5 mg) and stirred at 50 °C for 24 h. Thereafter, the reactions were filtered by using PTFE membrane, diam. 25 mm, pore size 0.22 μm, and freeze-dried. The crude was analyzed by gas chromatography associated to mass spectrometry (GC-MS) after traditional derivatization procedures.

### 4.4. Procedure for the Derivatization of the Sample

Samples (10.0 mg) were treated with an excess of BSTFA (N, N-bis-trimethylsilyl trifluoroacetamide) in pyridine (620 μL) at 90 °C for 3 h. The analysis was performed in the presence of betulinic acid [3β-hydroxy-20(29)-lupaene-28-oic acid] (1.0 mg) as an internal standard. After the derivatization procedure, we observed the presence of unsoluble material in low amount, probably due to the formation of high-molecular-weight HCN polymers (not quantified in our case).

### 4.5. Analysis of Reaction Products

GC-MS analytical program: initial temperature set at 100 °C, held for 2 min; then ramped at 10 °C/min to reach 280 °C, and held at 280 °C for 40 min. The yield of reaction products was calculated as micrograms of product per 1.0 mg of reaction crude. The structure of reaction products was assigned on the basis of the comparison of the molecular ions and fragmentation peaks with original commercial samples and with data stored in the appropriate software (National Institute of Standards and Technology bank of data) and, when necessary, by the co-addition method with authentic standards.

### 4.6. Polymerization of 3′,5′ cGMP

The polymerization of the cyclic nucleotide was performed in accordance with the method outlined in [40]. For each sample, 150 μL of 3′,5′ cGMP H**^+^** form (which had undergone neither evaporation nor precipitation during the preparation steps) was concentrated from the initial 1 mM concentration in Super Purity unbuffered water by evaporation in Savant SpeedVac Concentrator in Eppendorf plastic tubes and cooling mode until the desired dryness was achieved. The formation of a white semi-solid aggregate was used to judge the degree of dryness. The samples were stored at −20 °C until they could be suspended in situ, following the conditions described in the Results section. The reaction was stopped by quick freezing and ethanol precipitation by addition of 3 μL of glycogen (Thermo Scientific™ #R0561, Waltham, MA, USA, 20 μg/μL in water), 0.3 M (final volume) of sodium acetate pH 7.5, and three volumes of 96% ethanol.

### 4.7. 5′ Terminal Labeling of the RNA Oligomers Formed in the Polymerization Experiments

The samples were transferred to the laboratory and the oligomers formed were labeled with [γ-^32^P]ATP (adenosine triphosphate) by phosphorylation with polynucleotide kinase. Phosphorylation was carried out by adding 0.5 μL of T4 polynucleotide kinase PNK (EC 2.7.1.78, 10 U/μL, New England Biolabs, Ipswich, MA, USA; # M0201 L), and 2 μL of 10 × PNK buffer and 0.5 μL [γ-^32^P]ATP to the polymerization reaction mixture in 20 μL, followed by incubation at 37 °C for 30 min. This enzyme catalyzes the transfer and exchange of Pi from the γ-position of ATP to the 5′-OH terminus of polynucleotides, and the removal of 3′-phosphoryl group from 3′-phosphoryl polynucleotides. One unit is defined as the amount of enzyme catalyzing the production of 1 nmol of phosphate to the 5′-OH end of an oligonucleotide from [γ-^32^P] ATP in 30 min at 37 °C. This procedure typically provides a specific activity of 15,000 cpm (counts per minute)/pmol. The oligomers were then precipitated by addition of 0.3 M (final volume) of sodium acetate pH 7.5 and three volumes of 96% ethanol, kept overnight at −20 °C, centrifuged, washed once with a 70% ethanol/water mixture, and dehydrated (Savant SpeedVac Concentrator by Thermo Fisher Scientific, Waltham, MA, USA,13,000 rpm, 10 min, room temperature, environmental atmospheric pressure). The same procedure was employed for the terminal labeling of RNA markers.

### 4.8. Polyacrylamide Gel Electrophoresis

Samples (in pellet form) were suspended in 100% formamide and separated on 18% polyacrylamide gels (19:1 ratio acrylamide: N,N’-methylene-bis-acrylamide) containing 7 M urea, in 1 × TBE (Tris Borate Ethylenediaminetetraacetic acid) buffer.

### 4.9. Image Analysis

The gels were digitalized and the quantitative analysis of dot-blot signals was performed with the NIH ImageJ software (version 1.54K; https://imagej.net/ij/, access on: 17 October 2024). The image was inverted (Edit/Invert) so that the background pixel values are near zero, which is required for correct calculation of “integrated density”. The “integrated density” is the sum of the values of the pixels in the selection. The same area was outlined for every band/line and for the respective background. The subtraction of the IntDevband relative to the IntDevBackground is expressed in DLU (digital light units).

## Data Availability

The data are contained within the article and Appendix A.

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
