# Peer review of "The “Bagno dell’Acqua” Lake as a Novel Mars-like Analogue: Prebiotic Syntheses of PNA and RNA Building Blocks and Oligomers"

_ijms, 2025, doi:10.3390/ijms26146952_

Round 1
Reviewer 1 Report
Comments and Suggestions for Authors
The paper investigated the formation of bio-essential organic compounds catalyzed by microbialites in Martian analogous soil. The motivation is important in understanding potential chemical evolution on ancient Mars. However, microbialites contain bio-essential organic compounds. Thus, investigations of the formation of bio-essential organic compounds need special attention, such as using 13C-labeled starting materials. This experiment is not conducted with such attention, and thus there is no way to identify whether the detected compounds are products or contaminants. Further, the product identification is insufficient. Many isomers provide a similar fragmentation pattern to the target compound. Many experiments and analyses of meteorite samples show a larger abundance of isomers than the target compounds. The authors need to analyze the commercial standards to identify the retention time of the products.
Reviewer 2 Report
Comments and Suggestions for Authors
This manuscript describes plausible prebiotic syntheses under natural conditions, using an alkaline lake as a Martian analogue. In the first part of the manuscript, a mixture of ammonium formate, diaminomaleonitrile (DAMN), and several amino acids in a mineral aqueous solution—prepared using lake water and its corresponding microbialite—is explored as a basis for increasing organic molecular complexity. As a result, different ribonucleic acid (RNA) and peptide nucleic acid (PNA) building blocks, along with small peptides, are produced. Subsequently, the oligomerisation of guanosine monophosphate (GMP) is studied under the influence of the natural lake environment. The length of the oligomers depends on reaction time, pH, salinity, and sunlight. These results are of interest in the fields of prebiotic chemistry, the origin of life, and Martian exploration. Therefore, this work merits publication in IJMS after minor revision.
Comments:
- Please define the acronyms the first time they appear in the text. For example: RNA (line 90), GMP (line 95), PNA (line 98), AICN (line 143), and so on.
- A more detailed discussion of Scheme 1 would be beneficial:
- To the best of my knowledge, the isomerisation of DAMN to diaminofumaronitrile (DAFN) has been reported under light/UV conditions. Has the isomerisation from DAMN to DAFN also been reported upon heating?
- In Scheme 1, Ferus et al. is cited, but this reference is missing from the reference list.
- How is the methylation of compound 7 to yield compound 8 possible?
- Please expand Scheme 1 to provide a clearer understanding of the hypothetical pathways.
- Please revise Figures 2 and 3. The photographs of the polyacrylamide gels and the corresponding histograms should appear on the same page along with their respective figure captions. If this is not possible, it may be preferable to divide each figure into two separate parts—one for the gels and another for the histograms—resulting in a total of four figures instead of two.
- Expand the legend of Table 1 and specify that RT stands for "retention time." I assume this is expressed in minutes, in accordance with the chromatogram provided in the supplementary information. The same clarification should be made for Table S2.
- Please explain the rationale for using two gas chromatography (GC) systems: Varian GC410-320MS and TRACE GC-MS with a Thermo Finnigan Polaris Ion Trap.
Round 2
Reviewer 1 Report
Comments and Suggestions for Authors I read through all the replies to the review comments. This revision is not enough. Many control experiments with incubation of microbialites and the analyses of the extracts are essential to distinguish contamination and product. Regarding product identification, fragmentation is not enough to distinguish similar isomers. There are many possibilities for producing similar isomers in this experiment.Author Response
Please see the attachment

Round 3
Reviewer 1 Report
Comments and Suggestions for Authors
On the product identification, my question is simple. The authors should show LC-MS data to show the identification of products using identical retention time and identical fragmentation patterns. They just show the retention time and fragmentation patters on products. They should show all the retention time and identical fragmentation patterns of the commercial standards or in-house standards after identification by for example NMR. The author reply that the fragmentation pattern is sufficient to identify the compounds. However, even fragmentation patterns of the standard or NIST are not shown in the revised manuscript. How reviewers’ and authors understand that these fragmentation patters are identical to the shown products?
Further, for example, when we look at the composition of the orotic acid, C5H4N2O4, in ChemSpider, there are 133 isomers, some of the isomers are structurally very similar to orotic acid. How the authors confirmed that the fragmentation patterns of these isomers are not similar to orotic acid? Did the conformed with the fragmentation patterns in NIST? Some of the isomers are not common compounds. So, NIST does not cover all the compounds. Thus, the confirmation would be impossible. The simple solution I recommend is to check the identical retention time with standard. In the rebuttal letter, they are insisting that their identification is sufficient mentioning several previous papers in which the answer to my question is not explained. They do not reply to my question why they do not analyze standards.
Regarding the contamination, I’m satisfied with the author’s revision.
Author Response
We addressed the points in question by mail with the Editor.
Best Regards.
Bruno Mattia Bizzarri
Round 4
Reviewer 1 Report
Comments and Suggestions for Authors
I appreciate that the author understands my suggestion, and the quality of the manuscript has been improved. Many fragmentation spectra of standards differ significantly from those of the corresponding compounds after TMS derivatization. For example, the TMS derivative of orotic acid (compound 9) has fragments at m/z 254(100%), 100(40%), 147(40%), and 357(30%). However, the author's sample and standard orotic acid have m/z 73(100%), 213(50%), 102(40%). What makes this difference?
Finally, the author should describe the source of the commercial standard chemicals (manufacturer, quality) in the Method section, and the synthesis and identification of the synthesized standard chemicals with data in the Supplementary Information. Also, in the method section, the author should describe the details of GC-MS analysis, including the instrument, separation column, and temperature program. The description of "100 °C × 2 min" is not understandable.
